# Micropatterning Method for Porous Materials Using the Difference of the Glass Transition Temperature between Exposed and Unexposed Areas of a Thick-Photoresist

**DOI:** 10.3390/mi11010054

**Published:** 2019-12-31

**Authors:** Hidetaka Ueno, Kiichi Sato, Kou Yamada, Takaaki Suzuki

**Affiliations:** 1Division of Mechanical Science and Technology, Gunma University, Kiryu 376-8515, Japan; t162b001@gunma-u.ac.jp (H.U.); yamada@gunma-u.ac.jp (K.Y.); 2Division of Molecular Science, Gunma University, Kiryu 376-8515, Japan; kiichi.sato@gunma-u.ac.jp

**Keywords:** thick photoresist, glass transition temperature, backside exposure, scaffold, micropatterning

## Abstract

A cell culture on a scaffold has the advantages of functionality and easy handling, because the geometry of the cellular tissue is controlled by designing the scaffold. To create complex cellular tissue, scaffolds should be complex two-dimensional (2D) and three-dimensional (3D) structures. However, it is difficult to fabricate a scaffold with a 2D and 3D structure because the shape, size, and fabrication processes of a 2D structure in creating a cell layer, and a 3D structure containing cells, are different. In this research, we propose a micropatterning method for porous materials using the difference of the glass transition temperature between exposed and unexposed areas of a thick-photoresist. Since the proposed method does not require a vacuum, high temperature, or high voltage, it can be used for fabricating various structures with a wide range of scales, regardless of the materials used. Additionally, the patterning area can be fabricated accurately by photolithography. To evaluate the proposed method, a membrane integrated scaffold (MIS) with a 2D porous membrane and 3D porous material was fabricated. The MIS had a porous membrane with a pore size of 4 μm or less, which was impermeable to cells, and a porous material which was capable of containing cells. By seeding HUVECs and HeLa cells on each side of the MIS, the cellular tissue was formed with the designed geometry.

## 1. Introduction

An in vitro cell culture on a scaffold with microstructures can be used to create and elucidate biological functions and reproduction by constructing a biological structure similar to that which exists in vivo [1,2,3,4,5]. Cellular tissue for in vitro experiments is fabricated by spontaneous tissue formation. Since the external environment and structure affect spontaneous tissue formation, the geometry of the cellular tissue can be controlled by designing the scaffold for the cell culture. Additionally, since the cellular tissue is supported by the scaffold, it is more robust compared to natural cellular tissue or cellular tissue supported by gels. Scaffolding also enables easy handling, such as during transportation to another environment for cell culture or assay [6,7].

To elucidate the function of cellular tissue, it is necessary to create in vivo complex cell tissues in vitro. The in vivo cellular tissue consists of a 2-dimensional (2D) boundary, such as barrier tissues, and a 3-dimensional (3D) dense cell structure, such as organs. To reproduce these tissues by cell seeding on the scaffold alone, the scaffold also needs both 2D and 3D structures.

However, the fabrication methods for 2D shapes and 3D structures are often different, and these complex scaffold structures are difficult to produce by conventional fabrication methods. For example, the 2D microstructure is typically fabricated by photolithography or track-etching, which are suitable for a flat surface [8,9]. The 3D microstructure is typically fabricated by the phase separation method or 3D printing [10,11]. Moreover, it is difficult to integrate these micro-scale structures. For example, in the adhesion bonding method, the adhesive-filled microstructures create an unexpected adhesive layer due to surface tension. Additionally, solid bonding methods such as an anodic bonding are hard to apply to fragile materials such as porous materials, because it is limited to plane bonding and requires hard treatment conditions, such as a high temperature, high pressure, and high voltage [12].

In this study, we propose a micropatterning method on a porous material using the difference of the glass transition temperature between exposed and unexposed areas of a thick-photoresist. Since photoresists have been developed for manufacturing semiconductors and Micro Electro-Mechanical Systems (MEMS), the resolution of photoresists for the proposed method is enough to fabricate microstructures smaller than cell size. The microstructure is patterned on the porous material by partially reflowing the photoresist using the different glass transition temperatures of the exposed/unexposed areas. By using the proposed patterning method, a membrane integrated scaffold (MIS), with a 2D porous membrane capable of constructing cell boundaries, was patterned on a 3D porous material capable of containing cells internally. To evaluate the proposed MIS, two kinds of cells were respectively seeded on both sides of the fabricated MIS, and a co-culture test was performed.

## 2. Tissue on Artificial Scaffold

In vivo structures consist of 2D and 3D cellular tissues. 2D tissues mainly act as a barrier tissue; a boundary between the inside/outside of a living body, or between tissues [13]. For example, as shown in Figure 1a, vascular endothelial cells form a boundary by being arranged as the blood vessel wall, which is the 2D layer. Cells constituting an organ are the 3D cellular tissue attached to the blood vessel.

To fabricate cellular tissue in a shape similar to that found in vivo, the scaffold for fabricating the cellular tissue should also have 2D and 3D structures, such as a porous membrane and a porous material. Additionally, since in vitro cellular tissue studies require a large number of experiments, the devices need to be compatible with high reproducibility, mass production, and easy handling. An artificial scaffold with 2D and 3D structures has the potential to produce cellular tissue according to the scaffold structure by seeding cells (Figure 1b).

## 3. Materials and Methods

### 3.1. Principles of the Micropatterning Method

The micropatterning method for porous materials in this paper is composed of a partial reflow process dependent on the difference of the glass transition temperature between the exposed/unexposed areas of a thick-photoresist, and a releasing process using the sacrificial layer. The principle of the proposed method is shown in Figure 2. The method uses the porous material SU-8, which is a negative type thick-photoresist, a sacrificial layer for releasing, and two glass masks.

The characteristics of the exposed area of the SU-8, which is the negative type photoresist, are changed by cross-linking after the exposure process. A single SU-8 layer with thickness t_1_ is fabricated by the single spin-coating process. Three-dimensional distribution of exposure in the SU-8 layer is achieved by exposure energy UV_1_ with the first glass mask. In general, in a negative type photoresist, the exposed area is cross-linked and becomes insoluble during development, and the exposure depth correlates with the exposure energy. Therefore, the thickness ratio of the exposed area t_2_ and the unexposed area t_3_ is controlled by adjusting the first exposure energy UV_1_. By exposing the micropattern with exposure energy capable of exposing a depth of several micrometers t_2_, the patterning accuracy, depending on the aspect ratio, is kept high. Here, the cross-linked SU-8 changes its glass transition point to a higher temperature than the uncross-linked SU-8. Since the characteristics of SU-8 are affected by cross-linking, the glass transition temperature is different between the exposed/unexposed areas in a single SU-8 layer. Partial reflow, according to the distribution of exposure energy in the SU-8 layer, is generated by heating to a temperature which is higher than the glass transition temperature of the unexposed area and lower than that of the exposed area. The unexposed SU-8 layer, reflowed by heating, permeates into the porous material by capillarity. SU-8 permeating into the porous material is secondly exposed by exposure energy UV_2_ with the second glass mask. The second exposure energy UV_2_ is higher than the first exposure energy UV_1_, and is sufficient to expose the total film thickness t_1_. The second exposed area in the SU-8 becomes the adhesion area between the porous material and the micropattern. In addition, the area becomes a frame structure for the free-standing fabricated structures. The micropattern and adhesion area are fabricated by removing the unexposed area that has not been exposed to UV_1_ and UV_2_ by the single developing process.

The porous material with the micropattern is released by etching the sacrificial layer between the SU-8 layer and the first glass mask. Since the micropattern and porous material are bonded by the adhesion area, which is made of SU-8 permeating to depth *t_3_* into the porous material, the micropattern does not separate from the porous material. Therefore, the micropattern remains on the porous material.

### 3.2. Materials

In this paper, the SU-8 3005 (SU-8, MicroChem Corp., Westborough, MA, USA.), which is an epoxy-based photosensitive material, was used as the thick photoresist. Scaffold membrane AVP 004-48 (Alvetex scaffold, ReproCELL Inc., Kanagawa, Japan) was used as the porous material. These materials have been used for various biological experiments [14,15].

SU-8, the negative type photoresist, is capable of fabricating a layer with a thickness ranging from micrometer-scale to submillimeter-scale. The minimum pattern size of SU-8 is lower than 1 μm, which is smaller than cell size. Since SU-8 has not only a high resolution but also a high aspect ratio of over 5, as well as high chemical stability, it is used for fabricating 3D microstructures, as a mold for microchannels, and as a scaffold for cell cultures [16,17,18]. Normally, negative type photoresists are exposed after becoming solid by evaporating their solvent. SU-8 is capable of reflowing by heating, even once its solvent has evaporated. The glass transition temperature of SU-8 is 50–55 °C for the unexposed area, and 200 °C for the exposed area [19].

Alvetex scaffold is a porous material made of polystyrene for cell cultures. The porosity is 90% and it almost has a void structure. It is capable of containing cells because the pore size is larger than the cell size. The cells seeded on the Alvetex scaffold are attached to the surface and make cellular tissue inside it. Since the thickness of the Alvetex scaffold is 200 μm, the same as the distance of capillary blood vessels, there are no concerns regarding necrosis due to inhibiting circulation of nutrients and metabolized materials by the cells [20].

### 3.3. Process Flow of the Micropatterning Method

The MIS consists of a porous membrane with pores smaller than cells that is not permeable by the cells, and a porous material containing cells. The schematic images of the MIS are shown in Figure 3. The Alvetex scaffold consists of the porous material. The porous membrane adheres to the Alvetex scaffold at the adhesion area.

The fabrication process consists of eight steps. The process flow for fabricating the MIS is shown in Figure 4. First, the Cr layer was deposited on a cleaned glass substrate by sputtering (E-200S, Cannon Anelva, Kanagawa, Japan). Next, the pattern for the porous membrane was fabricated by photolithography and etching, using a positive type photosensitive resist Microposit S1813G (Rohm and Haas Electronic Materials, Marlborough, MA, USA) and the Cr etchant TK (Hayashi Pure Chemical Ind., Ltd., Osaka, Japan). The sacrificial layer, Omnicoat (MicroChem Co., Westborough, MA, USA.), which was made of the compound aliphatic imide, was spin-coated on the substrate at 1000 rpm and baked at 200 °C for 1 min. SU-8 3005 was spin-coated on the Omnicoat layer at 1000 rpm. To remove the solvent, the SU-8 layer was baked at 65 °C for 1 min and 95 °C for 1 min. The SU-8 layer was exposed from the backside of the substrate (PM50C-125A1, Ushio Inc., Tokyo, Japan). After the SU-8 layer was exposed, the Alvetex scaffold was put on the SU-8 layer without mechanical pressure. During the post exposure bake (PEB) at 65 °C for 1 min and 95 °C for 1 min, the unexposed SU-8 recovered flowability and permeated the Alvetex scaffold. The SU-8 was cooled down to room temperature. By using the mask with the pattern for the adhesion area, the SU-8 layer was exposed at 70 mJ/cm^2^ from the backside of the substrate. After PEB at 65 °C for 1 min and 95 °C for 1 min, the SU-8 was developed by the SU-8 developer (MicroChem Corp., Westborough, MA, USA.) and agitated by an ultrasonic wave. The SU-8 developer was replaced by isopropyl alcohol (Kanto Kagaku, Tokyo, Japan). Finally, the Alvetex scaffold integrated microstructure, made of SU-8, was released from the substrate by etching the Omnicoat layer using tetramethylammonium hydroxide (NMD-3, Tokyo Ohka Kogyo Co., Ltd., Kanagawa, Japan).

The surface of the fabricated MIS was observed by scanning electron microscope (SEM, JCM-5700LV, JEOL Ltd., Tokyo, Japan). The size of the micropattern was measured by image analysis, using Image J (National Institute of Health, MD, USA).

In the proposed method, the size of pores on the porous membrane was decided by the pattern size, which was on the glass substrate and the exposure energy. As parameters for finding out the minimum size of the pores we could fabricate, we prepared 4, 6 and 8 μm porous patterns on the glass substrate. Additionally, the parameter values of exposure energy were 30, 40, and 70 mJ/cm^2^.

### 3.4. Cell Culturing on the MIS

Cells seeded on the MIS were human umbilical vein endothelial cells (HUVECs) and HeLa-H2B-GFP cells (HeLa cells), which is the cell line of human cervix epithelioid carcinoma that has green fluorescence protein (GFP) in the core histone. Both cells are adherent cells derived from humans. The medium for each cell type was endothelial cell growth medium 2 (ECBM2, C-22111, Promo Cell. Inc, Heidelberg, Germany) and Dulbecco’s modified eagle’s medium, high glucose (DMEM, D5796500ML, Sigma, Mo, USA), to which we added 10 v/v% fetal bovine serum (35-076-CV, Corning, NY, USA) and 0.2 v/v% penicillin-streptomycin (P4333-20ML, Sigma, Mo, USA), respectively. Both cells were seeded in a petri dish and cultured. HeLa cells were peeled off from the petri dish and seeded on the Alvetex scaffold side of the MIS. After the HeLa cells attached and created cellular tissue, HUVECs were also peeled off from the petri dish and seeded on the porous membrane side of the MIS. Both cells were co-cultured for three days on the MIS, and observed by a fluorescence microscope.

As a preliminary treatment, the MIS was sterilized with ethanol and coated by a fibronectin. The fabricated MIS was sterilized with 78% ethanol. The 78% ethanol was then removed by phosphate buffered salts (PBS, T900, Roman Industries Co., Tokyo, Japan). The MIS was dipped into 10 mL PBS containing 1 μL fibronectin (FC010, Merck KGaA, Darmstadt, Germany) for 30 min. The excess fibronectin was removed by dipping in pure PBS three times. The silicone rubber sheet, with a thickness of 1.5 mm and ϕ4 mm hole, was set in the petri dish after it was sterilized with 78% ethanol. The MIS was placed on the silicone rubber sheet with the Alvetex scaffold side up. To lock the MIS, the poly dimethylpolysiloxane (PDMS) chip, the thickness of which was 3 mm with a ϕ4 mm hole, was set on the MIS after it was sterilized with 78% ethanol. The ϕ4 mm hole would work as a chamber for dropping cell suspension.

HeLa cells were cultured in a petri dish with a diameter of ϕ90 mm (I-90-20, BIO-BIK). After the HeLa cells became confluent, the DMEM was removed. The HeLa cells were washed with 10 mL PBS, then 1 mL of cell releasing solution (0.25% tripsin-EDTA(1X), Waltham, MA, USA) was dropped into the petri dish. After 5 min incubation, the cell suspension was created by adding 9 mL DMEM. After cell suspension, the mixture was moved into a centrifuge tube, and it was centrifuged at 900 rpm for 3 min by a centrifugal separator (SRX-201, TOMY SEIKO Co., Ltd., Tokyo, Japan). After removing the supernatant, 1 mL of DMEM was added, then 300 μL of the cell suspension was dropped on the Alvetex scaffold side of the MIS. After 2 mL of DMEM was added into the petri dish, the MIS was incubated for 24 h. After picking the MIS up from the silicone rubber sheet, the MIS was incubated for 20 days in another petri dish with 3.5 mL of DMEM.

HUVECs were cultured in a petri dish (MS-13900, Sumitomo Bakelite Co., Ltd., Tokyo, Japan) with a diameter of ϕ90 mm, until the area covered by cells reached 90%. The cell stain solution was fabricated by mixing 20 μL of dimethyl sulfoxide (DMSO, 317275, Sigma, Mo, USA) and 50 μg of fluorescence dye (CellTracker RED, Thermo Fisher Scientific, Waltham, MA, USA) with 0.2 v/v% penicillin-streptomycin (P4333-20ML, Sigma, Mo, USA) and 2 mL DMEM. After the petri dish was washed with 10 mL PBS, cell stain solution was added to the petri dish. After 30 min incubation, the cell stain solution was replaced by 10 mL PBS. After adding 5 mL ECBM2, it was incubated for 2 h. The ECBM2 was replaced by 10 mL PBS. After the PBS was removed, 1 mL of cell releasing solution was dropped into the petri dish. After 5 min incubation, the cell suspension was created by adding 9 mL of ECBM2. After the cell suspension was moved to a centrifuge tube, it was centrifuged at 900 rpm for 3 min by the centrifugal separator. After removing the supernatant, 1 mL ECBM2 was added, and 300 μL cell suspension was dropped on the porous membrane side of the MIS. After 2 mL ECBM2 was added into the petri dish, the MIS was incubated for 24 h. HUVECs and HeLa cells which did not adhere to the MIS were removed by washing the MIS in PBS. The MIS was put into a petri dish with ECBM2 and co-cultured for three days.

Co-cultured HUVECs and HeLa cells on the MIS were observed by fluorescence microscopy. One mL PBS was added into another petri dish. The MIS was placed into a petri dish with the HUVECs-seeded side down. The side on which the HUVECs were seeded was observed by fluorescence microscopy. After that, 1 mL PBS was added into another petri dish. The MIS was placed into a petri dish with the HeLa cells, seeded-side down. The side on which the HeLa cells were seeded was observed by fluorescence microscopy. Finally, the MIS was cut using tweezers, and its cross-section was observed by fluorescence microscopy. Two filter sets were used for fluorescence microscopy: filter set 1 (excitation filter 470–495 nm, emission filter 510–550 nm) and filter set 2 (excitation filter 545–580 nm, emission filter 610 nm).

## 4. Results

### 4.1. Micropatterning on Porous Material

The porous membrane patterned on the Alvetex scaffold was observed by SEM, and its size and fabrication error were calculated by image analysis. The size and relative fabrication error of the pores on the fabricated MIS are shown in Figure 5. The X-axis is the designed pore size. The Y-axis is the fabricated pore size. The R-axis is the relative fabrication error. The size of the pores is shown as circles. The relative fabrication error, which is the difference between the designed size and the fabricated pore size, is shown as triangles. Exposure was conducted with three conditions. Since there were no patterns when it was exposed at 30 mJ/cm^2^, these results are not plotted. The pattern size and the relative fabrication error of pores fabricated at 40 mJ/cm^2^ and 70 mJ/cm^2^ are shown by the fill pattern and blank pattern, respectively. The fabricated pore size after exposure at 40 mJ/cm^2^ was larger than after exposure at 70 mJ/cm^2^. This means that there is an inverse relationship between the fabricated pore size and the exposure energy. The relative fabrication error at 40 mJ/cm^2^ was smaller than at 70 mJ/cm^2^. This means there is a proportional relationship between the relative fabrication error and the exposure energy. To fabricate a pore smaller than 4 μm, the exposure energy must be 40 mJ/cm^2^ and the designed pore size must be almost 4 μm.

The fabricated MIS was observed by SEM. SEM images of the fabricated MIS are shown in Figure 6. The porous membrane side of the MIS is shown in Figure 6a. The porous membrane, the adhesion area, and the porous material were observed. The porous membrane adhered on the Alvetex scaffold. There was no damage on the other side, which was not patterned (Figure 6b). The size of the pores on the porous membrane was 3.46 ± 0.07 μm (Figure 6c). All pores on the porous membrane were opened. After cutting the porous membrane, the cross-section of the porous membrane tilted at 45°, as observed by SEM. The Alvetex scaffold was observed under the porous membrane. From the image analysis, we could see the thickness of the porous membrane was 8.29 ± 0.11 μm (Figure 6d).

### 4.2. Cell Culturing on the MIS

Cells on the MIS were observed by fluorescence microscopy. First, we observed the inside of the MIS, which was seeded with HeLa cells. After 4 days and 20 days, the MIS which was seeded with HeLa cells was cut, and its cross-section was observed. The fluorescence images are shown in Figure 7. At 4 days after seeding, the HeLa cells adhered on only the surface of the Alvetex scaffold. At 20 days after seeding, the HeLa cells proliferated and filled in the scaffold. Some HeLa cells reached the adhered side of the porous membrane.

After 21 days of seeding HeLa cells on the Alvetex scaffold side of the MIS, HUVECs were seeded on the porous membrane. After 3 days of co-culturing, fluorescence images of the porous membrane side, Alvetex scaffold side, and the cross-section of the MIS were taken. The fluorescence images are shown in Figure 8. Using filter set 1, HUVECs, which had red fluorescence dye, were observed on the porous membrane (Figure 8a). Using the Filter set 2, no cells with fluorescence were observed on the porous membrane (Figure 8b). Using filter set 1, no cells with fluorescence were observed on the Alvetex scaffold (Figure 8c). Using filter set 2, HeLa cells, which had green fluorescence, were observed on the Alvetex scaffold (Figure 8d). The MIS was cut using tweezers. The cross-section of the MIS was observed using filter set 1 and filter set 2. Fluorescence images were merged by ImageJ. The merged image is shown in Figure 8e. HUVECs were on the porous membrane. HeLa cells were not only on the surface of the Alvetex scaffold, but also inside it. Both cells were separated by the porous membrane.

## 5. Discussion

### 5.1. Comparison with Other Micropatterning Methods

In this paper, we proposed a micropatterning method for porous material. This involved patterning the porous membrane which had 3.5 μm pores and a thickness of 8.5 μm, to the porous material of the Alvetex scaffold which had over 50 μm porous by exposure energy at 40 mJ/cm^2^. The patterned porous membrane had pores smaller than the pore size of a typical porous material. In the patterning process for a porous material, the influence on the patterning by the surface roughness of the porous material should be considered. For example, using screen printing to fabricate a pattern on a porous material is difficult for a pattern that is smaller than the size of the pores on the porous material, because photosensitive material permeates into the pores on the porous material [21,22]. The proposed method fabricates fine patterns smaller than the pores on the porous material because the micropattern is fabricated on a smooth surface and integrated by being peeled off from the substrate. Additionally, since the patterned structure is not affected by porous material, the proposed method is able to be applied to various materials and shapes, not only to an Alvetex scaffold.

There was an inverse relationship between the pore size of the porous membrane and the exposure energy. Since SU-8 is a chemical amplification negative type resist, the pattern becomes larger in proportion to the exposure energy [16]. Therefore, the exposure energy should be minimized when fabricating smaller patterns. In the conventional body of research, a porous membrane with ϕ0.5 to 4 μm pores is fabricated by using a lower exposure energy of 20 mJ/cm^2^ [23]. Additionally, Esch et al. reported that a 0.5 to 2.5 mm thickness is necessary to hold the structure itself. However, in our research, the porous membrane was not fabricated at an exposure energy of 30 mJ/cm^2^. This is because, even though the exposure energy was enough, the external force applied to the porous membrane during the development and peeling process was too much for the porous membrane to resist. We used an ultrasonic cleaning process to mix the developer and developing SU-8 between the glass substrate and the Alvetex scaffold efficiently. Additionally, since the porous membrane was gradually separated from the glass substrate during the peeling process, a local external force was generated at the time of peeling. These external forces made the porous membrane fracture. Therefore, for fabricating thinner porous membranes with smaller pores, we have to consider the developing and peeling off processes, not just the exposure energy and pattern size. In the developing process in particular, SU-8 is developed without an ultrasonic cleaning process. A porous membrane which has smaller pores is able to fabricate with these changes in the fabrication process.

In this paper, the 2D porous membrane was fabricated on the 3D porous material for the border in cellular tissue. Some 2D porous membranes were used as the border for the cellular tissue alone. Basically, it is desirable that the pore size should be much smaller than the cell size, and the thickness of the porous membrane should be as small as possible. However, there is a trade-off relationship between the thickness and strength of the porous membrane. Therefore, these parameters should be decided by considering the requirements of the membrane, such as the seeded cell size. For example, in the microdevice used for fabricating barrier tissue in the lungs, a porous membrane with ϕ10 μm pores was used. In previous research on fabricating barrier tissue by using this porous membrane, the pore size and the thickness were decided based on the size of the cells cultured. The research used human lung alveolar epithelial cells and human lung capillary endothelial cells. These cells cannot pass through ϕ10 μm pores [24]. It was observed that only bacteria smaller than 10 μm went through cell layers on both sides of the porous membrane; the function of the barrier tissue was successfully reproduced. On the other hand, there are some cells that are smaller than 10 μm, such as granule cells [25]. For these cells, researchers are currently trying to fabricate a porous membrane with smaller pores that is very thin. Kim et al. fabricated ϕ0.8 μm and 1 μm thickness porous membranes [8]. In this study, HUVECs and HeLa cells did not pass through the fabricated porous membrane, as shown in Figure 8. Therefore, the fabricated pore size was small enough to seed these cells. Additionally, a 2D porous membrane integrated to the Alvetex scaffold was able to fabricate a physical border in the cellular tissue.

According to these results and this discussion, the proposed micropatterning method can be applied to various shapes and materials, such as porous materials. Additionally, by changing the conditions of the developing process, porous membranes suitable to other kinds of cells can also be fabricated.

### 5.2. Cellular Tissue Cultured on an MIS Made of Photoresist and Polystyrene

In this research, we chose a scaffold for cell cultures as one of the applications of the proposed patterning method. We fabricated the scaffold to have a 2D porous membrane for making the cell layer, and a 3D porous material for containing the cells. We cultured two kinds of adherent cells, HUVECs and HeLa cells, on each structure. Both cells were attached to each structure; the HeLa cells especially proliferated and filled up the Alvetex scaffold. The 2D porous membrane was made of SU-8, and the 3D porous material was the Alvetex scaffold, made of polystyrene. Since the Alvetex scaffold is normally used as a scaffold for cell cultures and polystyrene is a consistent material for common cell culture dishes, the 3D porous material was considered suitable for the cell culture. On the other hand, since SU-8 is photosensitive material, it is not as suitable for cell culture when compared with polystyrene. Surface treatment using a coating material or plasma treatment is needed to culture cells on the photoresist [14,26,27]. Since a porous membrane made of the photoresist is fragile, it is coated using poly D-lysine or collagen, and is not plasma treated [28,29]. In this research, we coated the surface of the SU-8 with fibronectin. Since the HUVECs were attached to the SU-8 surface coated by fibronectin, the shape of the cellular tissue was controlled by a structure made of SU-8. On the other hand, the metabolic function of the cells was elucidated by observing the shape of the cells or by measuring the metabolized material, such as albumin or urea [30]. In a future work, the metabolized function of the cellular tissue should be evaluated by comparing it with other cellular tissues cultured in different environments.

The autofluorescence of a photoresist impedes fluorescence microscopy if the photoresist is used as the scaffold for the cell culture. In particular, the signal-to-noise ratio of the fluorescence of the cells and scaffold is decreased. Image analysis cannot then be performed accurately. A photoresist with low autofluorescence was developed to solve this issue [31]. In a future work, the issue of the signal-to-noise ratio should be solved by using a photoresist with a higher resolution and lower autofluorescence, comparable to the SU-8.

According to these results and this discussion, a micropattern fabricated by patterning a photoresist can be used as a scaffold for cell cultures as long as a proper coating process is undertaken. In a future work, it is expected that a new method more suitable for cell cultures and fluorescence imaging could be developed by reducing the autofluorescence of the photoresist.

## 6. Conclusions

In this paper, we proposed a micropatterning method using the difference in the glass transition temperature between exposed and unexposed areas of a thick-photoresist. Using the proposed method, the 2D porous membrane, which has pores smaller than cells, is patterned on the 3D porous material, which can contain cells. Since the proposed method does not require a high temperature, high pressure, or high voltage, it can be used for a lot of different types of porous materials. Moreover, since two kinds of cells seeded on the MIS create the boundary inside the tissue, the MIS fabricated by the proposed method is capable of fabricating cellular tissue which has both a 2D and 3D shape, just by seeding cells.

## Figures and Tables

**Figure 1 micromachines-11-00054-f001:**
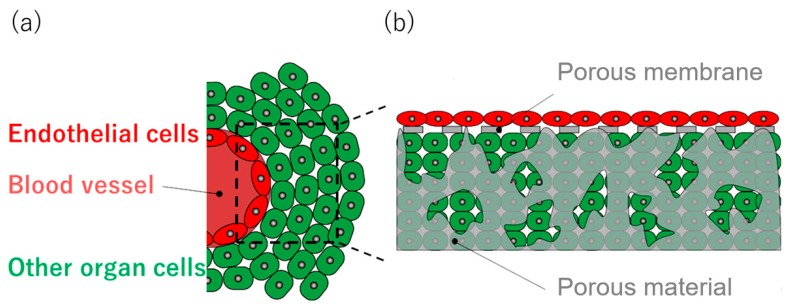
Schematic images of natural cellular tissue and cellular tissue on the scaffold. The cellular tissue inside the body consists of tissues with 2D/3D shapes. The cell layer which has a 2D shape creates a boundary between the inside/outside of the body, or between tissues. The 3D cellular tissue is arranged on the 2D tissue (**a**). The artificial scaffold MIS with a 2D porous membrane and the 3D porous material has the potential to create a similar structure to that of in vivo tissue, such as a blood vessel and organ cells, by seeding cells onto it (**b**).

**Figure 2 micromachines-11-00054-f002:**
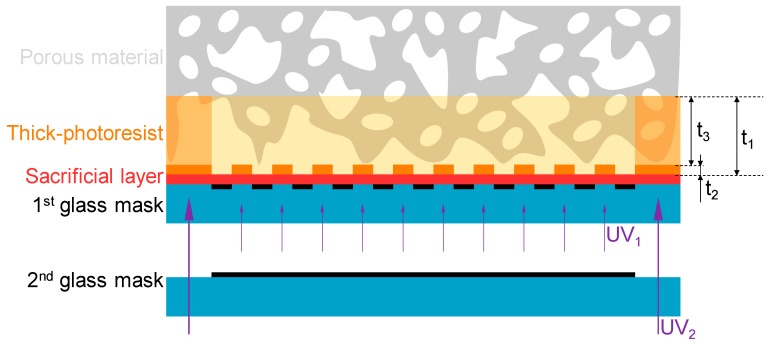
Principle of the proposed method. For micropatterning on a porous material, the method uses SU-8, which is a negative type thick photoresist, a sacrificial layer for releasing, and two glass masks.

**Figure 3 micromachines-11-00054-f003:**
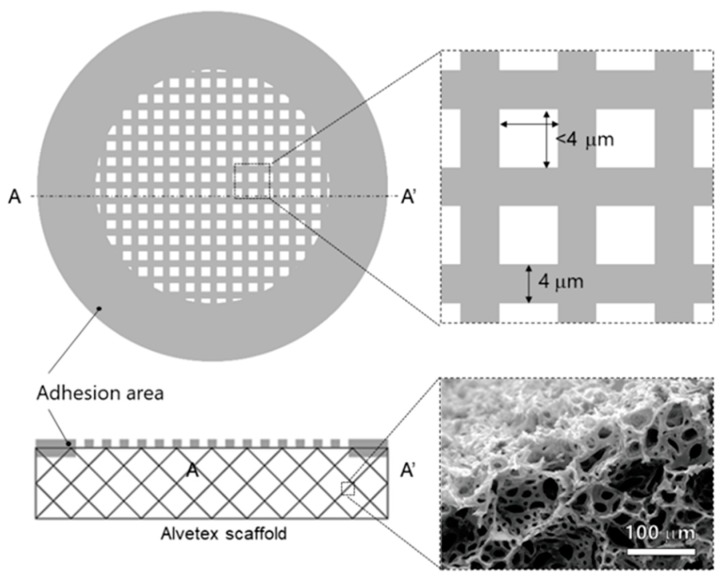
Schematic images of the MIS. The MIS consisted of the porous membrane with micro pore smaller than cell size, and the Alvetex scaffold which was capable of containing cells. The size of the pore in the porous membrane was required to be smaller than 4 μm. In the adhesion area, part of thick photoresist permeated into the Alvetex scaffold and adhered.

**Figure 4 micromachines-11-00054-f004:**
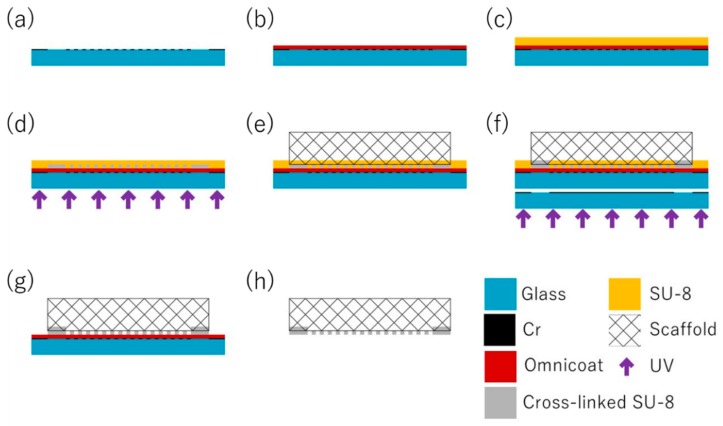
The fabrication process of the MIS. The MIS was fabricated by using the difference of the glass transition temperature between the exposed and unexposed areas of the thick-photoresist. First, the porous pattern was fabricated on the glass substrate by the Cr layer (**a**). The sacrificial layer was spin-coated on the substrate (**b**). SU-8 was spin-coated on the sacrificial layer (**c**). SU-8 was exposed from the backside of the substrate to the middle point of the SU-8 layer (**d**). The Alvetex scaffold was put on the SU-8 layer (**e**). The adhesion area was exposed from the backside of the substrate (**f**). The SU-8 layer was developed (**g**). Finally, the Alvetex scaffold adhered to the micropattern and was released from the substrate (**h**).

**Figure 5 micromachines-11-00054-f005:**
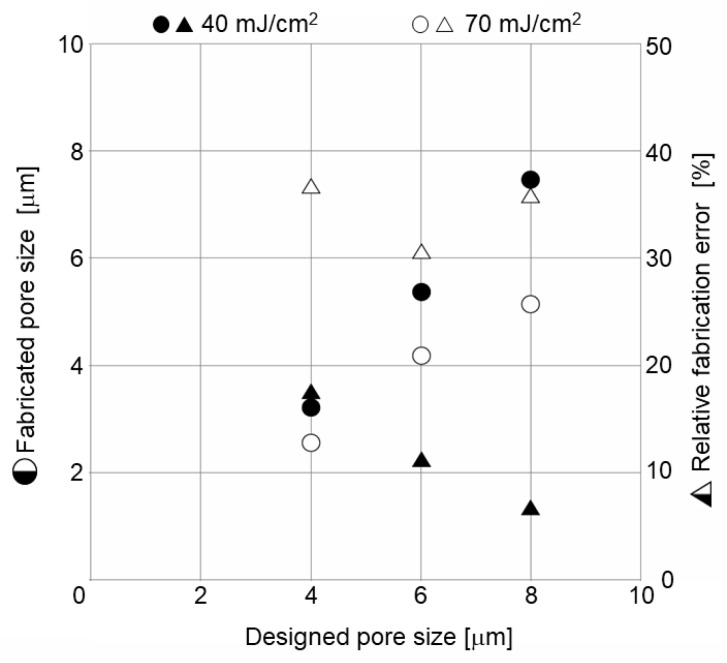
The size and the relative fabrication error of the pores on the fabricated MIS. There was no structure when it was fabricated with an exposure energy of 30 mJ/cm^2^.

**Figure 6 micromachines-11-00054-f006:**
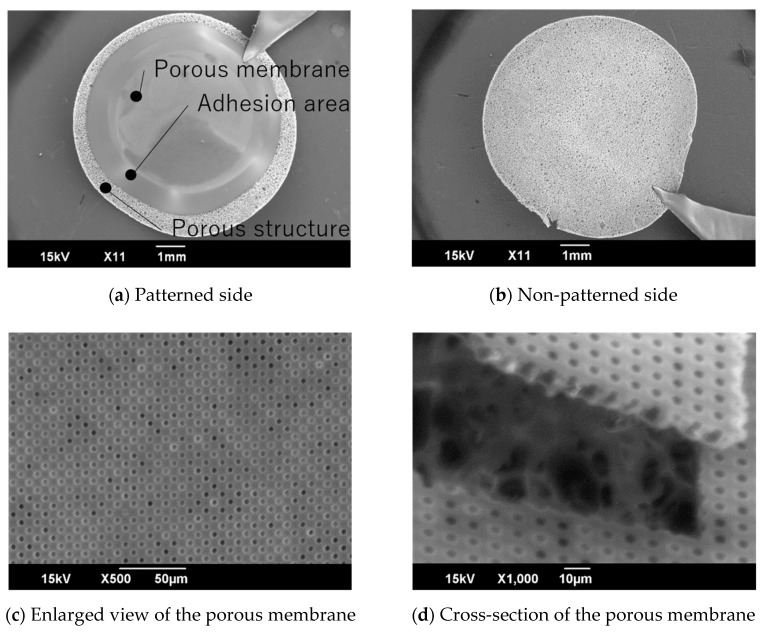
SEM images of the MIS. The porous membrane adhered on the Alvetex scaffold (**a**). There was no damage on the unpatterned surface of the Alvetex scaffold (**b**). All pores were opened on the porous membrane (**c**). The thickness of the porous membrane was about 8.3 μm (**d**).

**Figure 7 micromachines-11-00054-f007:**
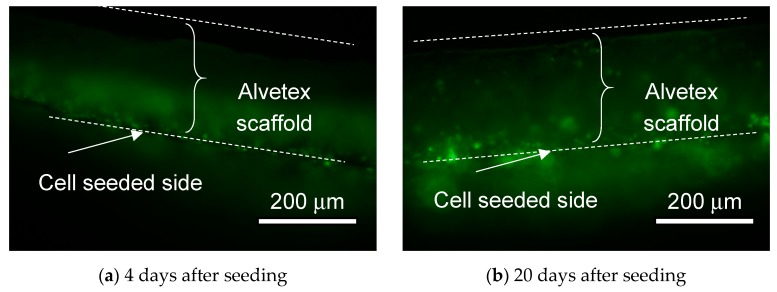
(**a**) HeLa cells attached to the surface of the Alvetex scaffold 4 days after seeding. (**b**) HeLa cells reached to the backside of the porous membrane 20 days after seeding.

**Figure 8 micromachines-11-00054-f008:**
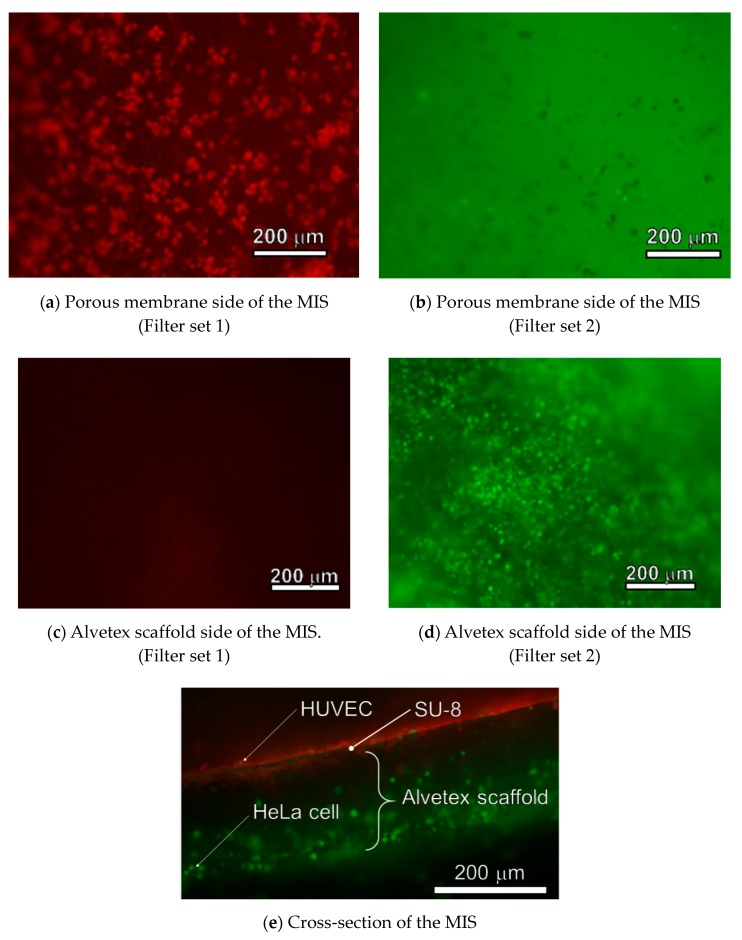
Fluorescence images of the MIS. HUVECs were observed on the porous membrane using filter set 1 (**a**). No cells were observed on the porous membrane using filter set 2 (**b**). No cells were observed on the Alvetex scaffold using filter set 1 (**c**). HeLa cells were observed on the Alvetex scaffold using filter set 2 (**d**). HUVECs, which were emitting red fluorescence, and HeLa cells, which were emitting green fluorescence, were separated by the porous membrane, made of the SU-8 (**e**).

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
