# Peer review of "Micropatterning Method for Porous Materials Using the Difference of the Glass Transition Temperature between Exposed and Unexposed Areas of a Thick-Photoresist"

_micromachines, 2019, doi:10.3390/mi11010054_

Round 1
Reviewer 1 Report
The paper entitled “Micropatterning method for porous materials by change of glass transition temperature of thick-photoresist” presents a novel microfabrication method that enables the creation of structures onto a porous material. The typical issue with creating such structures is the difficulty to create a strong adhesion between the two layers. The authors solved the issue by using typical photolithography but without dissolving the un-exposed photoresist. the un-exposed photoresist is left to flow into the porous substrate before being photo-polymerized to create a strong anchoring system.
The method developed is very interesting; however, the paper suffers greatly from the writing quality. At this point, it is even difficult to judge the results shown because the text needs extensive editing to address not only repeats but the whole text (grammar, …). In brief, the report could be very interesting, but it is so obscured by the writing that it is very difficult to review the scientific aspects.
Author Response
Thank you very much for providing valuable comments.
Major Comments:
Point 1: Extensive editing of the English language and style required.
Response 1: We have brushed this article up by the MDPI English calibration service.

Reviewer 2 Report
The manuscript by Ueno et al. demonstrates a method of fabricating porous membrane onto porous structure (Alvetex scaffold). The fabricated structure was used to coculture HUVEC and HeLa cells, with the former on the side of the fabricated porous membrane and the latter inside the Alvetex scaffold. This work is overall quite interesting as it is targeting at creating in vitro 3D hybrid tissue. The reviewer is pleased to recommend its publication if following concerns are addressed.
The glass transition temperature of the photoresist does not change. It is the difference of glass transition temperatures between exposed and unexposed photoresists that were used for the fabrication. The title and relevant descriptions should be revised accordingly. Some HeLa cells reached to the porous membrane side (SU-8 side) after 20 days (Line 234). In Line 293, HeLa cells were deemed incapable of passing through the pores. Please explain how the cells reached the other side of the membrane. Please swap part (c) and (d) in Figure 8. It is easier for the readers to understand if images of red and green channels are aligned in Figure 8. Figure 2 is redundant as the detailed fabrication process is present in Figure 4. Hence, Figure 2 may be deleted. Please provide catalog number and vendor of the porous structure Alvetex scaffold. What is the pore size of this material? Figure 8(a) shows that HUVEC cells were not densely grown on the porous membrane. Is this due to low seeding density or due to non-uniform coating of fibronectin? Line 310-311, combination of plasma treatment and coating material was also used previously to improve the coating. As an example, authors may refer to reference [Tu et al. "A microfluidic chip for cell patterning utilizing paired microwells and protein patterns." Micromachines 8.1 (2017): 1.] English needs considerable edits before publication.Author Response
Thank you very much for providing valuable comments.
Major Comments:
Point 1: Extensive editing of English language and style required.
Response 1: We have brushed this article up by the MDPI English calibration service.
Point 2: The glass transition temperature of the photoresist does not change. It is the difference of glass transition temperatures between exposed and unexposed photoresists that were used for the fabrication.
Response 2: We have changed the title of this article and the changed some sentences in this article. Changes sentences are shown in BLUE color.
Point 3: Some HeLa cells reached to the porous membrane side (SU-8 side) after 20 days (Line 234). In Line 293, HeLa cells were deemed incapable of passing through the pores. Please explain how the cells reached the other side of the membrane.
Response 3: HeLa cells were not pass through the pores. It reached to the backside which is the adhered side of porous membrane which is made of SU-8. We have changed the words from the backside to the adhered side in Line 259. The changed sentence is shown in RED color.
「Some HeLa cells reached to the backside of the porous membrane.」
➡「Some HeLa cells reached to the adhered side of the porous membrane.」
Point 4: Please swap part (c) and (d) in Figure 8. It is easier for the readers to understand if images of red and green channels are aligned in Figure 8.
Response 4: We have swapped these images and changed some sentences for explaining. Changed sentences are shown in RED color.
「By using the Filter set 2, there are no cells which had fluorescent were observed on the porous membrane (Figure 8(b)). By using the Filter set 2, HeLa cells which had green fluorescent were observed on the Alvetex scaffold (Figure 8(c)). By using the Filter set 1, there were no cells which had fluorescent were observed on the Alvetex scaffold (Figure 8(d)).」
➡「By using the Filter set 1, there were no cells which had fluorescent were observed on the Alvetex scaffold (Figure 8(c)).By using the Filter set 2, HeLa cells which had green fluorescent were observed on the Alvetex scaffold (Figure 8(d)).」
「HeLa cells were observed on the Alvetex scaffold by Filter set 2 (c). No cells were observed on the Alvetex scaffold by Filter set 1 (d).」
➡「No cells were observed on the Alvetex scaffold by Filter set 1 (c).HeLa cells were observed on the Alvetex scaffold by Filter set 2 (d).」
Point 5: Figure 2 is redundant as the detailed fabrication process is present in Figure 4. Hence, Figure 2 may be deleted.
Response 5: This research paper focuses to micro patterning method. So, we think that the section and figure for principle of the proposed method is necessary. Since the difference between principle (Fig.2) and process flow (Fig.4) was not clear, we have revised the figure.1, figure.2, and sentences. Changed sentences are shown in RED color.
Point 6: Please provide catalog number and vendor of the porous structure Alvetex scaffold. What is the pore size of this material?
Response 6: The catalog number and vendor of the porous structure Alvetex scaffold have been shown in Line 116. The size of pore is not constant because it is fabricated by phase separation method. However, the pore size is larger than cell size because this material is made for containing cells. I have write this point in the Line 127.
「Alvetex scaffold is a porous structure made of polystyrene for cell culture. The porosity is 90 % and almost void structure. It is capable of containing cells because the porous size is larger than cells.」
Point 7: Figure 8(a) shows that HUVEC cells were not densely grown on the porous membrane. Is this due to the low seeding density or due to non-uniform coating of fibronectin?
Response 7: This is due to the low seeding density. The reason why we did not seed cells with high density is for preventing necrosis. We used 300 mL cell suspension which is maximum volume can be dropped into chamber on the PDMS chip. If the cell density was too high, each cells did not obtain enough nutrients. We took fluoresce images after three days from seeding cells because the fluorescent dye keep its fluoresce in three days. At that time, there were some cells which is attached on the porous membrane. It means that the fabricated porous membrane is suitable to cell attaching. Also, it is expected that cells will proliferate on the porous membrane if we keep the cell culture. This is commented by Prof. Sato.
Point 8: Line 310-311, combination of plasma treatment and coating material was also used previously to improve the coating. As an example, authors may refer to reference [Tu et al. "A microfluidic chip for cell patterning utilizing paired microwells and protein patterns." Micromachines 8.1 (2017): 1.]
Response 8: We have added the research paper as the reference the number of which is 29.
Point 9: English needs considerable edits before publication.
Response 9: We requested native speakers of English to proofread our English writing.

Round 2
Reviewer 2 Report
Authors have addressed my previous concerns.